# Exposures to Potentially Psychologically Traumatic Events among Canadian Coast Guard and Conservation and Protection Officers

**DOI:** 10.3390/ijerph192215116

**Published:** 2022-11-16

**Authors:** Katie L. Andrews, Laleh Jamshidi, Jolan Nisbet, Taylor A. Teckchandani, Jill A. B. Price, Rosemary Ricciardelli, Gregory S. Anderson, R. Nicholas Carleton

**Affiliations:** 1Canadian Institute of Public Safety Research and Treatment (CIPSRT), University of Regina, Regina, SK S4S 0A2, Canada; 2Fisheries and Marine Institute, Memorial University of Newfoundland, St. John’s, NL A1C 5R3, Canada; 3Faculty of Science, Thompson Rivers University, Kamloops, BC V2C 0C8, Canada

**Keywords:** Public Service Personnel (PSP), Posttraumatic Stress Injury (PTSI), critical incident, mental disorders

## Abstract

Canadian Public Safety Personnel (PSP) (i.e., municipal/provincial police, firefighters, paramedics, Royal Canadian Mounted Police, correctional workers, dispatchers) report frequent and varied exposures to potentially psychologically traumatic events (PPTEs). Exposure to PPTEs may be one explanation for the symptoms of mental health disorders prevalent among PSP. The objective of the current study was to provide estimates of lifetime PPTE exposures among Canadian Coast Guard (CCG) and Conservation and Protection (C&P) Officers and to assess for associations between PPTEs, mental health disorders, and sociodemographic variables. Participants (*n* = 412; 55.3% male, 37.4% female) completed an online survey assessing self-reported PPTE exposures and self-reported symptoms of mental health disorders. Participants reported higher frequencies of lifetime exposures to PPTEs than the general population (all *ps* < 0.001) but lower frequencies than other Canadian PSP (*p* < 0.5). Several PPTE types were associated with increased odds of positive screens for posttraumatic stress disorder, major depressive disorder, general anxiety disorder, social anxiety disorder, panic disorder, and alcohol use disorder (all *ps* < 0.05). Experiencing a serious transportation accident (77.4%), a serious accident at work, home, or during recreational activity (69.7%), and physical assault (69.4%) were among the PPTEs most frequently reported by participants. The current results provide the first known information describing PPTE exposures of CCG and C&P members, supporting the growing evidence that PPTEs are more frequent and varied among PSP and can be associated with diverse mental health disorders.

## 1. Introduction

Public safety personnel (PSP) include, but are not limited to, border services officers, correctional workers, firefighters (career and volunteer), indigenous emergency managers, operational and intelligence personnel, paramedics, police (municipal and provincial), those responsible for public safety communication, Royal Canadian Mounted Police, and search and rescue personnel [1]. Within the Department of Fisheries and Oceans Canada (DFO), beyond the recognition that seafarers are inherently first responders in adverse events, at least two operating agencies include PSP, the Canadian Coast Guard (CCG) and Conservation and Protection (C&P) Officers.

The CCG helps to ensure Canada’s sovereignty and security by maintaining a presence in Canadian waters supported by more than 6100 employees. The CCG include PSP with duty-specific responsibilities that involve search and rescue operations within four regions: (1) Atlantic; (2) Central; (3) Arctic; and (4) Western regions. The CCG responds to about 6000 calls for marine assistance each year, with an average daily coordination of 19 search and rescue cases, resulting in the assistance of 68 individuals and, approximately, 18 lives saved [2]. The number of search and rescue incidents in Newfoundland and Labrador is estimated to be twice the national average [3]. C&P PSP have duty-specific responsibilities related to law enforcement and the protection of species at risk, fish habitat and oceans. There are more than 600 C&P personnel across the country, located in seven regions: (a) Newfoundland and Labrador; (b) Gulf; (c) Maritimes; (d) Quebec; (e) Arctic; (f) Ontario and Prairie; and (g) Pacific. C&P personnel are trained to carry out a wide range of duties, both on land and at sea, overtly and covertly. C&P personnel can encounter confrontational members of the public in remote locations, with little to no backup assistance.

PSP are exposed to hundreds or thousands of diverse potentially psychologically traumatic events (PPTEs) as a function of their services [4]. PPTEs include direct or indirect exposures to actual or threatened death, serious injury, or sexual violence [1]. Approximately 30% of the general population report never having been exposed to a PPTE and 40% report being exposed to between one and four PPTEs [5]. Most people who are exposed to PPTEs report being exposed to approximately two different types of PPTEs [6]. Exposure to PPTEs can be related to symptoms of posttraumatic stress disorder (PTSD) [7] and other mental health disorders (e.g., Major Depressive Disorder [MDD], Generalized Anxiety Disorder [GAD], and Social Anxiety Disorder [SAD]), collectively referred to as posttraumatic stress injuries (PTSIs) [1]. Accordingly, most of the general population experiences relatively few PPTEs and most do not develop a PTSI [8].

Recent research [4] has highlighted a high prevalence of mental health disorders and PPTE exposures among PSP. The PPTE exposures appear associated with PSP reporting PTSI symptom severity that greatly exceeds the general population [9,10]. The frequency of positive screens among PSP (44.5%) is much higher than the frequency of diagnosed mental health disorders in the general population [9]. Similar results were reported for provincial correctional workers (58.2% screened positive) [11] and police [12]. Additionally, recent research including other Canadian PSP (i.e., municipal/provincial police, firefighters, paramedics, Royal Canadian Mounted Police, correctional workers, dispatchers) reported positive screens for PTSD, MDD, and GAD to be significantly associated with almost all types of PPTEs reported by Canadian PSP [4,13] and nurses [14]. Like other PSP, CCG and C&P are expected to regularly face high-stress situations and multiple exposures to PPTEs as part of their daily work activities. Consequently, researchers have suggested that CCG and C&P members may experience mental health disorders with a prevalence higher than the general population [5,6].

Information regarding CCG and C&P members’ mental health and associated factors, such as the prevalence of PPTE exposure for CCG and C&P members, is limited, and relies heavily on research conducted with members of the United States Coast Guard (USCG). The limited available research has reported 15% of USCG met the diagnostic criteria for PTSD and 5% met the diagnostic criteria for MDD [15]. Results from the 2018 Department of Defense Health Related Behaviors Survey (HRBS) [16] indicate that symptoms of psychological distress (10.6%) and PTSD (7.3%) in the past year were common among members of the USCG. Members also reported exposure to unwanted sexual contact (10.0%) and physical assault (3.3%) since joining the military [16]. The 2018 HRBS results may be relevant to the CCG and C&P due to the overlap and long-standing cooperation between the organizations. Both services regularly share information and equipment, exchange personnel, conduct joint training exercises, and rely on each other to save lives [17]. Despite the similarities and shared duties, the CCG and C&P have yet to be included in recent research on Canadian PSP duty-related mental health issues.

There is currently no published research regarding PPTE exposures among CCG and C&P members. There is also no published research regarding the relationship between PPTE exposures and mental health disorders among CCG and C&P members. Understanding the experiences of CCG and C&P members may provide important insights that can inform training efforts to protect their mental health and extend their years of service. Previous research with PSP evidenced a specific subset of PPTE as being consistently the “worst” and most likely to be associated with mental health disorders, suggesting additional resources need to be available at the discretion of individual PSP [4]. No such assessments have been made regarding CCG and C&P experiences.

The current study was designed to: (1) assess the history and prevalence of PPTE exposures among CCG and C&P members; (2) clarify the PPTEs perceived by CCG and C&P members as the worst events; (3) assess relationships between lifetime PPTE exposures and positive screening for diverse mental health disorders; and (4) compare PPTE exposure experiences across demographic categories. PPTE exposure frequencies for CCG and C&P members were expected to be higher than the general population, evidenced across previous PSP research [4]. Furthermore, per prior PSP research [4,13,14], PPTE exposures were expected to be associated with positive screens for mental health disorders.

## 2. Materials and Methods

### 2.1. Procedure

Data were collected using a web-based self-report survey in English or French. The study was approved by the University of Regina Institutional Research Ethics Board (REB# 2021-003). The survey was based on a set of validated measures used in a previous study of PSP [4,9,10,11,12,13,14], but collaboratively redesigned by the research team and the CCG and DFO teams to ensure relevant variables were included. The survey was promoted and distributed by the CCG and DFO to member unions via emails, social media posts, and a video encouraging participation. The survey was available between 1 February 2021, to 31 January 2022. Participants were provided with a randomly generated unique code upon entering the survey which allowed for repeated survey access to complete the survey over multiple sessions. The current study focused specifically on the self-reported symptoms of mental health disorders and exposures to PPTEs.

### 2.2. Data and Sample

Participants were CCG/DFO members (*n* = 412) (67.5% CCG members and 26.0% C&P members). Responses from 561 CCG/DFO members were initially collected. Only data from respondents who completed at least 30% of the survey were retained. The final sample included in the current study analyses and results had a total of 412 members. Participants were mainly male (55.3%), identifying as men (55.3%), 30 to 39 years old (26.9%) or 40 to 49 years old (26.5%), with a college (37.4%) or university (31.3%) degree (see Table 1). Participants were mostly white (82.8%), married or in common-law relationships (63.8%), and resided in British Columbia (53.2%), with no previous experience in the Canadian Armed Forces or as PSP (67.5%).

### 2.3. Self-Report Measures

The survey included the Life Events Checklist for the Diagnostic and Statistical Manual of Mental Disorders 5-Extended (DSM-5) (LEC-5) [18]. Of note, the LEC-5 does not include the unexpected death of a loved one, an adverse event that no longer meets criteria for PTSD in the DSM-5 [7]. Participants reported on the PPTE exposure modality (e.g., indirectly or directly) and all their reported experiences were treated as exposures for the current article: (a) it happened to them personally; (b) they witnessed it happen to someone else; (c) they learned about it happening to a close family member or close friend; and/or (d) they were exposed to it as part of their job. The total number of different PPTE exposure types was quantified by summing exposure frequencies across the 17 items. The LEC-5 Extended was modified to ask participants to report the number of exposures to each PPTE type they reported being exposed to. If the participant reported exposure to more than one PPTE type they were asked to select the worst PPTE or the PPTE currently causing them the most distress, as well as the number of exposures to that PPTE type, and the length of time since the first and the last (most recent) exposure.

The survey also asked participants to self-report symptoms related to various mental disorders. A ‘positive screen’ on any of the measures indicated that the individual self-reported symptoms consistent with expectations for a diagnosis of a particular mental health disorder. A positive screen on a self-report survey is not necessarily synonymous with meeting diagnostic criteria, which requires a clinical interview by a licensed professional. However, substantial differences in rates when comparing self-reported mental disorder symptoms consistent with a positive screen and interview assessments were not identified in a recent meta-analysis [19]. Nonetheless, individuals who completed the self-report measures and indicated a positive screen would require the evaluation of a trained and licensed clinician for the possible diagnosis of a specific mental health disorder. The current study assessed symptoms related to screening positive for the mental disorders of PTSD, MDD, GAD, SAD, Panic Disorder (PD), and Alcohol Use Disorder (AUD).

The PTSD Checklist for DSM-5 (PCL-5) [20,21] assesses for symptoms related to PTSD. Participants rated how bothered they had been by 20 common symptoms of PTSD in the past month on a five-point Likert scale, from 0 (i.e., not at all) to 4 (i.e., extremely). Participants reported their behaviors over the past month. For the PCL-5, a positive screen required participants to report exposure to at least one LEC-5 item, meet minimum DSM-5 criteria for each PTSD symptom cluster subscale (e.g., intrusions, avoidance, negative alterations in cognitions and mood, and alterations in arousal and reactivity), and exceed the clinical cut-off of >32 [21]. Psychometric evaluation of the PCL-5 has demonstrated strong internal consistency (α = 0.94) and good test–retest reliability (r = 0.82) within populations exposed to PPTEs [20].

The nine-item Patient Health Questionnaire (PHQ-9) [22] assesses symptoms of MDD. Participants indicated how bothered they had been by depressive symptoms in the past two weeks by responding to each item using a four-point Likert scale (i.e., 0 = not at all to 3 = nearly every day). For the PHQ-9, a positive screen is indicated by a score of >9 [9]. Psychometric evaluation of the PHQ-9 has demonstrated good internal consistency (α = 0.89) and test–retest reliability (r = 0.84) within the general population [22].

The Panic Disorder Symptoms Severity Scale, Self-Report (PDSS-SR) [23] assesses symptoms of PD. Participants first read the definition of a panic attack, and the accompanying symptoms. From the accompanying symptoms, at least four had to be endorsed (e.g., rapid or pounding heartbeat, sweating, nausea, feeling of choking) for a panic attack to have occurred. If the participant reported having ever experienced a panic attack, or experiencing a panic attack in the past week, they were asked additional questions rated on a five-point Likert scale (i.e., 0 = none to 4 = extreme). A positive screen required the PDSS-SR total scores to be >7 [23]. The self-report version has demonstrated excellent psychometrics with a strong internal consistency (α = 0.92) and an intraclass correlation coefficient of 0.81 [24].

The seven-item Generalized Anxiety Disorder Scale (GAD-7) [25] assesses symptoms of GAD. Participants indicated the extent to which seven symptoms of anxiety bothered them in the previous two weeks. Ratings were made on a four-point Likert Scale (i.e., 0 = not at all to 3 = nearly every day). A positive screen for GAD required the GAD-7 total score to be >9 [25]. The GAD-7 has shown good reliability, and construct, criterion, procedural, and factorial validity [25], as well as good internal consistency (α = 0.89) and inter-item correlations 0.45–0.65 in a community sample [26].

The fourteen-item Social Interaction Phobia Scale (SIPS) assesses for symptoms of SAD [26]. The SIPS includes three subscales to assess social interaction anxiety, fear of overt evaluation, and fear of attracting attention, respectively. Each item is rated on a five-point Likert Scale (i.e., 0 = not at all characteristic of me, 4 = entirely characteristic of me). There is no specific time window used. A positive screen for SAD requires the SIPS total score to be >20 [27]. The SIPS has demonstrated overall excellent internal consistency (α = 0.92), convergent and discriminant validity in a large and independent sample [28].

The ten-item Alcohol Use Disorders Identification Test (AUDIT) assesses for symptoms of AUD [29]. Participants were asked questions about their drinking behaviors and negative alcohol-related consequences. Ratings were made using Likert scales that varied across items. A positive screen for AUD required the total AUDIT score to be >15 [30]. Psychometric evaluation of the AUDIT has demonstrated good internal consistency (α = 0.81), good test–retest reliability (r = 0.83 to 0.95) within the general population, and (α = 0.81) in a police-specific population [31,32,33].

### 2.4. Statistical Analyses

Participants were grouped into sociodemographic categories for comparisons on total number of different types of PPTEs. Descriptive analyses, including frequencies and percentages of the sociodemographic variables (i.e., sex, gender, age, education, ethnicity, marital status, province of work, previous work experience), and means and standard deviations of total number of different types of PPTEs were performed. A series of *t*-tests and one-way analysis of variance (ANOVA) tests were performed to compare total number of different types of PPTEs across sociodemographic categories for the total sample and for CCG and C&P samples, separately. All tests were two-tailed with an alpha level of 0.05. Holm–Bonferroni adjustments were applied to alpha levels in post-hoc tests to reduce familywise error rate. Prevalence rates of lifetime exposure to each PPTE, of the worst PPTE and the frequency of exposure (i.e., 1 to 5 times, 6 to 10 times, 11 times or more) were calculated for the total sample, as well as CCG and C&P samples. Prevalence of lifetime exposure to each PPTE were then compared against previously published data from a sample of other Canadian PSP (i.e., municipal/provincial police, firefighters, paramedics, Royal Canadian Mounted Police, correctional workers, dispatchers) [4] and general population [5,6], performing a series of binomial tests. Multivariate logistic regression models were conducted to assess the association between each type of traumatic exposure and positive screens for mental disorders for the total sample (CCG and C&P), and for separate CCG and C&P samples. All regression models were adjusted for sociodemographic covariates (i.e., sex, age, education, marital status, province, job category). All data were analyzed using SPSS v.28 Premium (IBM, 2021, New York, NY, USA).

## 3. Results

Sociodemographic characteristics are presented in Table 1. Males reported statistically significant higher total numbers of different types of PPTEs than females (effect sizes (*d*s) ranged from 0.245 to 0.451 across total sample, CCG, and C&P; all *ps* < 0.05). Statistically significant effects of province of work (effect sizes ($\eta_{p}^{2}$) ranged from 0.068 to 0.080 across total sample and CCG; all *ps* < 0.001) and previous work experience (effect sizes ($\eta_{p}^{2}$) ranged from 0.049 to 0.146 across total sample and C&P; all *ps* < 0.001) on total number of different types of PPTEs were observed. However, follow-up multiple pairwise comparisons were not statistically significant, due to the application of Holm–Bonferroni adjustment to alpha levels in post-hoc tests to control familywise error rate, except the C&P sample, wherein those with previous work experience as public safety workers reported a statistically significant higher total number of different types of PPTEs than those having no previous work experience as public safety workers or CAF (see Table 2).

Table 3 presents PPTE exposure prevalence for the total sample, and CCG and C&P samples, compared to previously published results of other Canadian PSP [4] and the general population [5,6]. The most frequently reported PPTE types were serious transportation accident (77.4%), serious accident at work, home, or during recreational activity (69.7%), physical assault (69.4%), sudden accidental death (69.4%), and fire or explosion (68.9%). Participants reported being exposed to PPTEs 1–5 times (9.2%), 6–10 times (11.6%), or 11+ times (78.9%). Participants reported statistically significant fewer PPTE exposures for all events than serving PSP (all *ps* < 0.001) [4], except for “combat” with statistically significant more exposure rate (*p* < *0*.05). Participants from the total sample reported statistically significant more PPTE exposures than the general population (all *ps* < 0.001) [4], except for “serious injury, harm, or death you caused to someone else”, where the general population reported more exposures (*p* < 0.001) [5,6]. Participants also reported statistically significant less types of PPTE exposures than other PSP and more types of PPTE exposures than the general population (all *ps* < 0.001).

Table 4 presents the prevalence of each PPTE type most frequently identified as the worst. Participants most frequently reported the worst event was a “serious transportation accident” (10.7%), “sudden violent death” (10.0%), or “life threatening illness or injury” (9.5%).

Associations between PPTEs and positive screens for mental health disorders among the total sample, and CCG, and C&P samples are presented in Table 5, Table 6 and Table 7. All logistic regression models were adjusted for covariates (i.e., sex, age, education, marital status, province, and job category). Statistically significant associations were observed between different PPTE types and increased odds of positive screens for PTSD, MDD, GAD, SAD, PD, and AUD across the total sample (all *ps* < 0.05). Positive screening for PTSD was statistically significantly associated with exposure to toxic substances, assault with a weapon, sexual assault, other unwanted or uncomfortable sexual experiences, and captivity (AORs ranged from 1.13 to 3.07, all *ps* < 0.05). Positive screening for MDD was statistically significantly associated with exposure to physical assault and captivity (AORs = 2.05 and 2.51, respectively, all *ps* < 0.05). Positive screening for GAD was statistically significantly associated with exposure to a toxic substance, captivity, and serious injury, harm, or death you caused to someone else (AORs range from 1.96 to 2.57, all *ps* < 0.05). Positive screening for SAD was statistically significantly associated with captivity (AOR = 2.43, *p* < 0.05). Positive screening for PDSS was statistically significantly associated with life threatening natural disaster, fire or explosion, captivity, and serious injury, harm, or death you caused to someone else (AORs range from 2.77 to 4.01, all *ps* < 0.05). Positive screening for AUD was statistically significantly associated with life threatening illness or injury (AOR = 0.28, *p* < 0.01).

## 4. Discussion

To the best of our knowledge, the current article presents the first evidence of PPTE exposures of CCG and C&P members and may generalize to other national coast guard groups and PSP populations. Participants were expected to report PPTE exposures greater than the general population, wherein 30% of people report never being exposed to a PPTE [5,6]. In contrast, only 7.5% of the current sample reported never being exposed to a PPTE. Most participants (92.5%) reported exposure to at least one PPTE. Participants in the total, CCG and C&P samples reported significantly more PPTE exposures compared to the general population and significantly fewer PPTE exposures compared to the other Canadian PSP [4]. CCG and C&P members reported exposure to an average of eight different types of PPTEs, with exposure frequencies of 1–5 times (9.2%), 6–10 times (11.6%), or 11+ times (78.9%). The results were consistent with expectations; specifically, CCG and C&P members experience many more PPTEs than the general population.

The most frequently reported PPTEs in the general population include sudden accidental death (41.1%), fire or explosion (33.2%), life threatening illness or injury (32.0%), severe human suffering (30.5%), and sudden violent death (28.2%) [5,6]. The most frequently reported PPTEs for the total sample were serious transportation accident (77.4%), serious accident at work, home, or during recreational activity (69.7%), physical assault (69.4%), sudden accidental death (69.4%), and fire or explosion (68.9%). Similar PPTEs were reported by the general population and CCG/C&P members, but the rates of exposure were higher for CCG and C&P members. The same PPTE types were also commonly reported among Canadian PSP [4], along with exposure to sudden violent death and assault with a weapon. The frequencies of exposure to each PPTE were significantly higher for other Canadian PSP [4] than CCG and C&P members. C&P members also reported life threatening disasters (74.8%) and assault with a weapon (74.8%). C&P members may engage in duties specific to law enforcement and protection of species at risk [34,35]; as such, C&P members may directly interact with the individuals who are hunting or fishing and likely to have weapons for those purposes. The differences in PPTE exposure frequencies and types between the general population, Canadian PSP, and the current sample, support the contention that PSP are exposed to a diverse range of PPTEs more frequently than the general population [4,34]. Differences in occupational duties may highlight types of PPTEs more likely to be encountered for specific PSP groups.

The PPTEs most frequently selected as the worst event by CCG and C&P members were serious transportation accident (10.7%), sudden violent death (10.0%), and life-threatening illness or injury (9.5%). CCG members reported serious transport accident as their worst PPTE, while C&P members reported sudden violent death as their worst PPTE. The group differences could be due to differences in duties between CCG and C&P members. The current results were consistent with previous research indicating the worst PPTEs reported by other Canadian PSP [4], which were also sudden violent death (28.0%) and serious transportation accident (13.9%). Sudden violent death (19.2%) and life-threatening illness or injury (10.7%) were also reported as the worst PPTEs by Canadian correctional workers [13]. The relative frequency of such PPTE types, coupled with the associated severity, may increase the probability of selecting such PPTE types as the worst event if asked to select a single event during a PTSD screening following the DSM-5 criteria for PTSD [7]. The relative frequency and variable severity of PPTEs experienced by CCG and C&P members also suggests that regular psychoeducation or evidence informed training, such as Emotional Resilience Skills Training (ERST) [36], or other mental health training [37], may be beneficial for protecting members’ mental health.

The current results further support associations between PPTE exposures and higher odds of screening positively for several different mental disorders. PPTE exposures were associated with screening positive for PTSD, MDD, GAD, SAD, PD, and AUD. Several PPTEs were associated with screening positive for PTSD, including assault with a weapon, sexual assault and other unwanted or uncomfortable sexual experience, and captivity. Captivity was associated with screening positive for most mental disorders including PTSD, MDD, GAD, SAD, and PD. The association could be due to differences in occupational demands which increases the probability of specific PPTE exposures. The associated mental health disorders and PPTEs were consistent between CCG and C&P samples, except that C&P reported increased odds for screening positive for SAD following several PPTEs. The results should be interpreted cautiously because of how few members screened positive for any mental health disorder, despite the prevalence of PPTE exposures. In any case, the current results further support arguments that, instead of any specific subset of PPTE types, a wide variety of PPTE types may be associated with PTSIs, including PTSD [4,13,14].

Very few differences in exposure frequencies were observed across sociodemographic categories. Some sex differences were observed. Males reported a significantly higher total number of different types of PPTEs than females. This was consistent with previous research examining PPTE exposures of Canadian PSP [4], RCMP cadets [38], and nurses [14]. Males reported more frequent exposures to serious transport accidents, whereas females reported more frequent exposures to other unwanted or uncomfortable sexual experiences. Only 37.4% of the sample were female, which suggests that the most frequent PPTE types across all CCG and C&P members might become other unwanted or uncomfortable sexual experiences, rather than serious transport accidents, as the relative proportion of female participants increases.

### Strengths and Limitations

The current study used data provided by a national and diverse sample of CCG and C&P members; however, several limitations caveat the current results and provide directions for future research. First, the survey was promoted and distributed by the CCG and DFO to personnel via emails, internal newsletters, internal webpages, videos, and posters encouraging participation. The CCG and C&P included approximately 6700 members (i.e., CCG 6100, C&P 600). The current sample reflected approximately 6.15% of CCG and C&P members and included larger proportions of CCG members (67.5%) than C&P members (26.0%). The current sample also included relatively larger proportions of members from British Columbia, and smaller proportions of members from Quebec, Nova Scotia, Newfoundland, and Labrador. Therefore, the current sample may not be entirely representative of the entire CCG and C&P workforce.

Second, participation in the current study was anonymous, voluntary, and self-selected. The recruitment materials described the study as focusing on PPTE exposures, mental health disorders, occupational stress, and burnout, which may have attracted participants who were experiencing clinically significant mental health symptoms. The recruitment materials may partially explain differences in prevalence rates between the current sample and the general population; however, CCG and C&P members who were experiencing the most severe symptoms may have been on leave, missed the invitation, or been too symptomatic to respond to a lengthy survey. Stigmatizing attitudes about mental health may also have inhibited some individuals from accessing the survey, despite assurances of anonymity. Additionally, the collection method, using an online survey, may have impacted the number of participants. Many CCG and C&P members do not have easy access to computers or internet as they serve on ships, in stations, or in the field and are often away for long periods of time. Participants were able to begin, leave, and return to the survey at their leisure, to ease the survey response burden; as such, there was no way to know the average length of survey completion time or to understand why some participants (6.5%) did not complete the entire survey.

Third, the screening measures for mental health disorders used in the current study are valid and reliable for use in clinical settings; nevertheless, diagnoses can only be made using clinical interviews with supporting collateral information. Participants reported on their current symptoms as assessed by the screening measures, with time periods ranging from seven days to the past year, and, specifically, reported on lifetime suicidal behaviors. Further, only a relatively small number of potential mental disorders were screened for in the current study. The frequency of positive mental health disorder screens lends support to the need for additional research using Statistics Canada sampling methods and clinical interviews to make more reliable assessments and to allow for comparisons with the general population. Clinical interviews assessing lifetime prevalence would also help to discern whether symptoms developed prior to, or over the course of, participants’ careers.

Despite the limitations, the survey demographics indicated the sample was generally proportionally consistent with the age and sex of CCG and C&P members. The current study was the first known national attempt to understand the impact of PPTE exposures and symptoms of mental health disorders among CCG and C&P members. The selected measures allowed for comparisons with other large occupational studies designed to estimate PPTE exposures in specific occupational samples, such as PSP [4,13,14]. The current results provide potentially important information to support researchers investigating possible ways to mitigate and manage PTSIs among PSP.

## 5. Conclusions

The current results offer the first known empirical evidence of PPTE exposure histories of CCG and C&P members. The results indicate that CCG and C&P members are exposed to diverse PPTEs more frequently than the general population, but less frequently than Canadian PSP. The results also indicate that PPTEs can be associated with mental health disorders. CCG and C&P may experience different PPTEs than other Canadian PSP due to differences in job activities. Rates of exposure may also be lower than other Canadian PSP, due to working in different, more isolated, environments. Differences in exposures to specific PPTE types was observed between males and females, which suggests that the most frequent PPTE types across all CCG and C&P members might differ as the relative proportion of female participants increases. The associations between PPTE exposures and mental disorders suggest a need for ongoing mental health training/support to mitigate the impact of service on mental health, particularly suited to remote or isolated locations.

## Figures and Tables

**Table 1 ijerph-19-15116-t001:** Sociodemographic Characteristics.

	Demographic Distribution % (*n*)		
	Total Sample	CCG	C&P
Total Sample	100 (412)	100 (278)	100 (107)
Gender			
Men	55.3 (228)	56.1 (156)	66.4 (71)
Women	36.7 (151)	41.7 (116)	31.8 (34)
Non-Binary	1.2 (5)	^	^
Indigenous-Two-Spirits	-	-	-
Sex			
Male	56.1 (231)	56.8 (158)	67.3 (72)
Female	37.4 (154)	42.4 (118)	32.7 (35)
Age			
19–29	11.9 (49)	12.2 (34)	13.1 (14)
30–39	26.9 (111)	30.2 (84)	25.2 (27)
40–49	26.5 (109)	25.9 (72)	33.6 (36)
50–59	22.3 (92)	23.4 (65)	25.2 (27)
60+	5.1 (21)	6.8 (19)	^
Education			
High School or Less	8.5 (35)	10.8 (30)	4.7 (5)
College Program (e.g., Trade School; 2-Year College Diploma)	37.4 (154)	36.3 (101)	49.5 (53)
Coast Guard College: Graduated Fleet	9.5 (39)	14.0 (39)	-
Coast Guard College: MCTS Officer Training	2.2 (9)	3.2 (9)	-
University Degree (4-year College or Higher)	31.3 (129)	28.8 (80)	43.9 (47)
Ethnicity			
Asian	2.2 (9)	^	4.7 (5)
Black	^	^	-
Hispanic	^	-	^
Indigenous (i.e., First Nations, Inuit, Métis)	3.2 (13)	2.5 (7)	5.6 (6)
South Asian	^	^	-
White	82.8 (341)	89.6 (249)	85.0 (91)
Prefer Not to Answer	1.2 (5)	^	^
Other	3.6 (15)	4.3 (12)	^
Marital Status			
Married/Common Law	63.8 (263)	65.5 (182)	75.7 (81)
Single	20.6 (85)	22.7 (63)	18.7 (20)
Separated/Divorced	7.3 (30)	9.0 (25)	4.7 (5)
Widowed	^	^	-
Province of Work			
Alberta	-	-	-
British Columbia	53.2 (219)	53.6 (149)	63.6 (68)
Manitoba	-	-	-
New Brunswick	1.7 (7)	-	6.5 (7)
Newfoundland and Labrador	6.8 (28)	7.6 (21)	6.5 (7)
Northern Territories (YK, NWT, NVT)	^	^	-
Nova Scotia	9.0 (37)	7.9 (22)	14.0 (15)
Ontario	10.9 (45)	15.5 (43)	^
Prince Edward Island	-	-	-
Québec	11.9 (49)	15.1 (42)	6.5 (7)
Saskatchewan	-	-	-
Previous Work Experience			
Neither	67.5 (278)	80.2 (223)	49.5 (53)
CAF Only	8.3 (34)	10.1 (28)	5.6 (6)
Public Safety Only	15.8 (65)	7.6 (21)	41.1 (44)
CAF and Public Safety	2.4 (10)	2.2 (6)	^

Note. -: *n* = 0; ^: Sample size between 1 and 5, so data not presented; Lettered superscripts within each column category indicate significant differences between category groups with different letters on outcome at *p* ≤ 0.05. CAF = Canadian Armed Forces; CCG = Canadian Coast Guard; C&P = Conservation and Protection; NWT = Northwest Territories; NVT = Nunavut; YK = Yukon. Total percentages may not sum to 100 and *n*s may not sum to 412, due to non-response.

**Table 2 ijerph-19-15116-t002:** Total Number of Different Types of Potentially Psychologically Traumatic Events (PPTEs) Exposures Across Total Sample, CCG, and C&P Officers.

	Total Number of Different Types of PPTEs					
	Total Sample	Test Results	CCG	Test Results	C&P	Test Results
	M(SD)	Effect Size	M(SD)	Effect Size	M(SD)	Effect Size
Total Sample	8.72 (4.42)	-	9.05 (3.87)	-	9.76 (4.07)	-
Gender						
Men	9.71 (3.87)	0.029 *	9.40 (3.82)	0.027	10.32 (3.89)	0.062
Women	8.54 (3.98)		8.46 (3.90)		8.65 (4.23)	
Non-Binary	9.40 (4.16)		^		^	
Indigenous-Two-Spirits	-		-		-	
Sex						
Male	9.74 (3.86)	0.306 **	9.42 (3.80)	0.245 *	10.35 (3.90)	0.451 *
Female	8.54 (3.98)		8.48 (3.90)		8.54 (4.21)	
Age						
19–29	7.94 (4.32)	0.023	7.26 (4.27)	0.031	9.07 (4.03)	0.094 *
30–39	9.18 (3.79)		9.35 (3.70)		8.67 (4.10)	
40–49	9.82 (3.84)		9.43 (3.66)		10.44 (4.10)	
50–59	9.66 (3.87)		9.12 (3.98)		10.96 (3.31)	
60+	8.90 (4.23)		9.42 (3.98)		^	
Education						
High School or Less	8.51 (4.10)	0.008	8.80 (3.84)	0.010	6.80 (5.63)	0.026
College Program (e.g., Trade School; 2-Year College Diploma)	9.06 (3.83)		8.67 (3.88)		9.81 (3.66)	
Coast Guard College: Graduated Fleet	9.69 (3.19)		9.69 (3.19)		-	
Coast Guard College: MCTS Officer Training	7.89 (3.44)		7.89 (3.44)		-	
University Degree (4-year College or Higher)	9.38 (4.30)		8.94 (4.21)		9.89 (4.35)	
Ethnicity						
Asian	10.22 (3.77)	0.031	^	0.013	11.40 (3.05)	0.082
Black	^		^		-	
Hispanic	^		-		^	
Indigenous (i.e., First Nations, Inuit, Métis)	10.15 (4.51)		9.71 (5.02)		10.67 (4.23)	
South Asian	^		^		-	
White	9.11 (3.94)		8.94 (3.88)		9.49 (4.06)	
Prefer Not to Answer	13.20 (1.92)		^		^	
Other	10.87 (3.36)		10.08 (3.23)		^	
Marital Status						
Married/Common Law	9.16 (3.97)	0.014	8.82 (3.83)	0.020	9.93 (4.18)	0.021
Single	9.00 (4.09)		8.89 (4.07)		8.75 (4.00)	
Separated/Divorced	10.40 (3.40)		10.20 (3.61)		11.40 (2.07)	
Widowed	^		^		-	
Province of Work						
Alberta	-	0.068 ***	-	0.080 ***	-	0.047
British Columbia	9.95 (3.77)		9.80 (3.65)		10.13 (3.99)	
Manitoba	-		-		-	
New Brunswick	9.57 (4.93)		-		9.57 (4.93)	
Newfoundland and Labrador	8.89 (3.36)		9.24 (3.18)		7.86 (3.93)	
Northern Territories (YK, NWT, NVT)	^		^		-	
Nova Scotia	9.05 (4.50)		8.18 (4.69)		10.33 (4.03)	
Ontario	9.18 (4.02)		9.14 (3.90)		^	
Prince Edward Island	-		-		-	
Québec	6.84 (3.56)		6.76 (3.65)		7.29 (3.20)	
Saskatchewan	-		-		-	
Previous Work Experience						
Neither	8.77 (3.87)	0.049 ***	8.83 (3.82)	0.028	8.30 (4.00) ^b^	0.146 ***
CAF Only	10.65 (3.93)		10.64 (3.92)		10.67 (4.37) ^a,b^	
Public Safety Only	10.26 (3.95)		8.71 (4.24)		11.00 (3.62) ^a^	
CAF and Public Safety	12.40 (2.46)		11.33 (2.34)		^	

Note. -: *n* = 0; ^: Sample size between 1 and 5, so data not presented; Lettered superscripts(a,b) within each column category indicate significant differences between category groups with different letters on outcome at *p* ≤ 0.05. CAF = Canadian Armed Forces; CCG = Canadian Coast Guard; C&P = Conservation and Protection; NWT = Northwest Territories; NVT = Nunavut; YK = Yukon. * *p* < 0.05, ** *p* < 0.01, *** *p* < 0.001—Statistically significantly different at these *p*-value levels; Holm-Bonferroni adjustment applied to alpha levels to control Type I errors.

**Table 3 ijerph-19-15116-t003:** Prevalence of Potentially Psychologically Traumatic Events (PPTEs) Exposure Types.

Type of Exposure	Total Sample	CCG	C&P	PSP [4]	Comparing Total Sample and PSP	General Population [7]	Comparing Total Sample and General Population [7]	General Population [6]	Comparing Total Sample and General Population [6]
	% (*n*)	% (*n*)	% (*n*)	% (*n*)	Test Statistics	%	Test Statistics	%	Test Statistics
Life threatening natural disaster (e.g., flood, hurricane, tornado, earthquake, and wildfires)									
Ever exposed ^1^	65.0 (268)	66.9 (186)	74.8 (80)	66.4 (2832)	−0.53	15.6	27.59 ***	7.7	43.57 ***
1 to 5 times	86.1 (211)	88.4 (153)	80.6 (58)	91.7 (2376)					
6 to 10 times	6.5 (16)	5.8 (10)	8.3 (6)	3.5 (90)					
11 or more times	7.3 (18)	5.8 (10)	11.1 (8)	4.8 (125)					
Fire or explosion				-					
Ever exposed	68.9 (284)	73.4 (204)	72.9 (78)	86.0 (3727)	−9.91 ***	n/a	-	n/a	-
1 to 5 times	83.4 (216)	84.4 (157)	80.8 (59)	50.4 (1805)					
6 to 10 times	5.0 (13)	5.9 (11)	^	10.1 (361)					
11 or more times	11.6 (30)	9.7 (18)	16.4 (12)	39.6 (1419)					
Serious transportation accident (e.g., car accident, boat accident, train wreck, plane crash)				-					
Ever exposed	77.4 (319)	81.3 (226)	85.0 (91)	93.2 (4084)	−12.62 ***	17.8	31.58 ***	14.0	37.03 ***
1 to 5 times	68.0 (210)	65.3 (143)	74.4 (67)	22.7 (907)					
6 to 10 times	7.4 (23)	5.9 (13)	11.1 (10)	6.0 (241)					
11 or more times	24.6 (76)	28.8 (63)	14.4 (13)	71.3 (2845)					
Serious accident at work, home, or during recreational activity				-					
Ever exposed	69.7 (287)	73.4 (204)	75.7 (81)	81.6 (3430)	−6.19 ***	7.9	46.38 ***	6.2	53.31 ***
1 to 5 times	86.6 (232)	86.5 (167)	86.7 (65)	66.6 (2091)					
6 to 10 times	5.2 (14)	5.2 (10)	^	6.5 (205)					
11 or more times	8.2 (22)	8.3 (16)	8.0 (6)	26.9 (844)					
Exposure to toxic substance (e.g., dangerous chemicals, radiation)				-					
Ever exposed	50.0 (206)	53.2 (148)	52.3 (56)	67.4 (2664)	−7.48 ***	10.1	26.80 ***	4.2	46.22 ***
1 to 5 times	71.7 (132)	68.6 (94)	80.9 (38)	61.3 (1439)					
6 to 10 times	4.9 (9)	5.8 (8)	^	7.4 (174)					
11 or more times	23.4 (43)	25.5 (35)	17.0 (8)	31.3 (735)					
Physical assault (e.g., being attacked, hit, slapped, kicked, beaten up)				-					
Ever exposed	69.4 (286)	69.8 (194)	84.1 (90)	90.6 (3931)	−14.65 ***	9.2	42.20 ***	5.9	54.61 ***
1 to 5 times	75.0 (189)	76.0 (46.8)	72.8 (59)	41.8 (1543)					
6 to 10 times	8.7 (22)	7.6 (13)	11.1 (9)	9.5 (350)					
11 or more times	16.3 (41)	16.4 (28)	16.0 (13)	48.7 (1797)					
Assault with a weapon (e.g., being shot, stabbed, threatened with a knife, gun, bomb)				-					
Ever exposed	52.7 (217)	48.6 (135)	74.8 (80)	83.9 (3639)	−17.18 ***	16.0	20.24 ***	14.5	21.93 ***
1 to 5 times	92.2 (153)	94.1 (95)	89.2 (58)	57.6 (1797)					
6 to 10 times	^	^	^	9.4 (294)					
11 or more times	6.0 (10)	^	9.2 (6)	32.9 (1027)					
Sexual assault (e.g., rape, attempted rape, made to perform any type of sexual act through force or threat of harm)				-					
Ever exposed	52.4 (216)	55.8 (155)	55.1 (59)	71.2 (3035)	−8.52 ***	11.4	26.13 ***	5.8	40.39 ***
1 to 5 times	91.7 (143)	91.8 (112)	91.2 (31)	47.1 (1089)					
6 to 10 times	^	^	^	11.0 (255)					
11 or more times	5.8 (9)	4.9 (6)	-	41.9 (968)					
Other unwanted or uncomfortable sexual experience				-					
Ever exposed	59.2 (244)	64.7 (180)	57.9 (62)	67.3 (2803)	−3.44 ***	21.9	18.26 ***	4.2	55.56 ***
1 to 5 times	72.0 (152)	72.5 (116)	72.0 (36)	55.2 (1225)					
6 to 10 times	5.2 (11)	5.0 (8)	^	7.7 (171)					
11 or more times	22.7 (48)	22.5 (36)	22.0 (11)	37.2 (825)					
Combat				-					
Ever exposed	22.6 (93)	21.9 (61)	28.0 (30)	18.8 (791)	1.90 *	4.3	18.16 ***	3.2	22.20 ***
1 to 5 times	73.1 (38)	71.4 (25)	76.5 (13)	78.4 (349)					
6 to 10 times	^	^	^	21.6 (96)					
11 or more times	23.1 (12)	22.9 (8)	-	-					
Captivity (e.g., being kidnapped, abducted, held hostage, prisoner of war)				-					
Ever exposed	15.5 (64)	14.7 (41)	19.6 (21)	30.5 (1279)	−6.55 ***	1.4	24.21 ***	1.1	27.85 ***
1 to 5 times	92.5 (37)	96.2 (25)	85.7 (12)	78.9 (712)					
6 times or more	-	-	^	21.2 (191)					
11 or more times	^	^	-	-					
Life threatening illness or injury				-					
Ever exposed	66.5 (274)	69.8 (194)	72.9 (78)	76.7 (3301)	−4.84 ***	32.0	14.96 ***	11.8	34.34 ***
1 to 5 times	83.4 (206)	84.4 (152)	80.3 (53)	54.2 (1594)					
6 times or more	5.3 (13)	4.4 (8)	7.6 (5)	6.3 (184)					
11 or more times	11.3 (28)	11.1 (20)	12.1 (8)	39.6 (1165)					
Severe human suffering				-					
Ever exposed	53.2 (219)	56.5 (157)	56.1 (60)	79.1 (3234)	−12.89 ***	3.4	55.59 ***	3.7	53.05 ***
1 to 5 times	70.8 (126)	66.4 (87)	82.6 (38)	41.8 (1187)					
6 times or more	7.3 (13)	9.2 (12)	^	6.2 (177)					
11 or more times	21.9 (39)	24.4 (32)	15.2 (7)	51.9 (1473)					
Sudden violent death (e.g., homicide, suicide)				-					
Ever exposed	65.8 (271)	69.1 (192)	72.0 (77)	93.8 (4101)	−23.49 ***	n/a	-	n/a	-
1 to 5 times	80.9 (195)	78.8 (134)	85.9 (61)	36.4 (1426)					
6 times or more	6.6 (16)	5.9 (10)	8.5 (6)	13.1 (512)					
11 or more times	12.4 (30)	15.3 (26)	^	50.5 (1977)					
Sudden accidental death				-					
Ever exposed	69.4 (286)	73.4 (204)	74.8 (80)	93.7 (4063)	−20.19 ***	41.1	11.63 ***	1.4	117.30 ***
1 to 5 times	78.5 (194)	76.1 (137)	85.1 (57)	34.6 (1321)					
6 times or more	6.9 (17)	7.2 (13)	^	10.7 (408)					
11 or more times	14.6 (36)	16.7 (30)	9.0 (6)	54.8 (2092)					
Serious injury, harm, or death you caused to someone else				-					
Ever exposed	13.8 (57)	12.9 (36)	19.6 (21)	36.2 (1485)	−9.40 ***	n/a	-	26.3	−5.69 ***
1 to 5 times	80.0 (28)	81.0 (17)	78.6 (11)	64.4 (580)					
6 times or more	^	-	^	6.8 (61)					
11 or more times	17.1 (6)	^	^	28.9 (260)					
Any other very stressful event or experience				-					
Ever exposed	31.1 (128)	30.6 (85)	39.3 (42)	-	-	8.6	16.18 ***	40.8	−3.97 ***
1 to 5 times	72.1 (98)	69.7 (62)	78.3 (36)	-					
6 times or more	8.1 (11)	9.0 (8)	^	-					
11 or more times	19.9 (27)	21.3 (19)	15.2 (7)	-					
Total number of different types of potentially traumatic exposures, *Mean* (*SD*)	8.72 (4.42)	9.05 (3.87)	9.76 (4.07)	11.08 (3.23)	t (411) = −10.85 ***	2.31 (2.33)	t (411) = 29.41 ***	2.2-3.7	-

Notes: CCG = Canadian Coast Guard; C&P = Conservation and Protection; PSP = public safety personnel; *SD* = Standard Deviation; n/a = not available. ^1^ Not all individuals ever being exposed reported the number of exposure. So, the number of exposures does not necessarily match ever exposed. ^ Sample size between 1 and 5, so data not presented. Expanded table available in Appendix A. * *p* < 0.05, *** *p* < 0.001—Statistically significantly different at these *p*-value levels.

**Table 4 ijerph-19-15116-t004:** Prevalence of Worst Potentially Psychologically Traumatic Events (PPTEs) Exposure Types CCG, C&P, and Previously Published PSP.

Type of Worst Exposure	Total Sample	CCG	C&P	PSP [4]
	% (*n*)	% (*n*)	% (*n*)	% (*n*)
Life threatening natural disaster (e.g., flood, hurricane, tornado, earthquake, and wildfires)	1.5 (6)	^	^	2.0 (77)
Fire or explosion	2.7 (11)	2.9 (8)	^	3.2 (123)
Serious transportation accident (e.g., car accident, boat accident, train wreck, plane crash)	10.7 (44)	11.9 (33)	10.3 (11)	13.9 (540)
Serious accident at work, home, or during recreational activity	3.9 (16)	4.7 (13)	^	3.4 (130)
Exposure to toxic substance (e.g., dangerous chemicals, radiation)	^	^	-	0.5 (18)
Physical assault (e.g., being attacked, hit, slapped, kicked, beaten up)	4.9 (20)	4.3 (12)	7.5 (8)	4.9 (190)
Assault with a weapon (e.g., being shot, stabbed, threatened with a knife, gun, bomb)	2.4 (10)	^	5.6 (6)	6.3 (245)
Sexual assault (e.g., rape, attempted rape, made to perform any type of sexual act through force or threat of harm)	5.3 (22)	6.5 (16)	^	5.1 (196)
Other unwanted or uncomfortable sexual experience	3.6 (15)	4.3 (12)	^	1.4 (53)
Combat	^	^	^	1.1 (43)
Captivity (e.g., being kidnapped, abducted, held hostage, prisoner of war)	^	^	-	0.6 (25)
Life threatening illness or injury	9.5 (39)	10.8 (30)	8.4 (9)	6.6 (255)
Severe human suffering	5.3 (22)	5.4 (15)	6.5 (7)	7.0 (272)
Sudden violent death (e.g., homicide, suicide)	10.0 (41)	10.1 (28)	12.1 (13)	28.0 (1086)
Sudden accidental death	8.0 (33)	9.0 (25)	7.5 (8)	14.0 (542)
Serious injury, harm, or death you caused to someone else	1.9 (8)	1.8 (5)	^	2.1 (8)
Any other very stressful event or experience	15.0 (62)	14.7 (41)	18.7 (20)	-

Notes: ^ Sample size between 1 and 5, so data not presented. CCG = Canadian Coast Guard; C&P = Conservation and Protection; PSP = Public Safety Personnel.

**Table 5 ijerph-19-15116-t005:** Associations Between Potentially Psychologically Traumatic Events (PPTEs) Exposure and Screening Positive for Mental Disorder (Adjusted for Sociodemographic and Job Categories).

Type of PPTE	PTSD	MDD	GAD	SAD	PD	AUD
	AOR (95% CI)	AOR (95% CI)	AOR (95% CI)	AOR (95% CI)	AOR (95% CI)	AOR (95% CI)
Life threatening natural disaster	1.34 (0.65, 2.79)	1.23 (0.68, 2.23)	0.89 (0.44, 1.80)	1.33 (0.70, 2.55)	3.75 (1.04, 13.55) *	0.36 (0.14, 0.92)
Fire or explosion	1.63 (0.74, 3.57)	1.29 (0.70, 2.40)	1.61 (0.73, 3.58)	1.19 (0.61, 2.34)	3.94 (1.02, 15.27) *	0.60 (0.22, 1.62)
Serious transportation accident	1.59 (0.60, 4.20)	0.65 (0.32, 1.31)	1.19 (0.48, 2.97)	0.86 (0.40, 1.84)	1.24 (0.36, 4.26)	1.07 (0.31, 3.71)
Serious accident at work, home, or during recreational activity	1.33 (0.61, 2.90)	1.02 (0.55, 1.91)	1.18 (0.55, 2.51)	0.78 (0.41, 1.49)	2.06 (0.63, 6.71)	0.76 (0.29, 1.98)
Exposure to toxic substance	2.88 (1.44, 5.77) **	1.43 (0.83, 2.44)	1.96 (1.01, 3.83) *	1.46 (0.80, 2.64)	2.01 (0.82, 4.95)	0.91 (0.37, 2.24)
Physical assault	2.19 (0.94, 5.10)	2.05 (1.05, 4.01) *	1.47 (0.68, 3.19)	1.11 (0.56, 2.20)	1.86 (0.63, 5.52)	0.80 (0.29, 2.15)
Assault with a weapon	2.24 (1.10, 4.58) *	1.34 (0.76, 2.37)	0.95 (0.49, 1.85)	1.01 (0.55, 1.88)	1.64 (0.64, 4.17)	0.49 (0.19, 1.27)
Sexual assault	1.97 (1.01, 3.84) *	0.86 (0.51, 1.45)	0.96 (0.51, 1.83)	0.92 (0.51, 1.66)	1.18 (0.50, 2.81)	0.75 (0.30, 1.84)
Other unwanted or uncomfortable sexual experience	2.26 (1.09, 4.69) *	1.24 (0.71, 2.18)	0.87 (0.44, 1.71)	1.28 (0.68, 2.43)	0.87 (0.35, 2.14)	1.05 (0.39, 2.83)
Combat	2.01 (0.99, 4.10)	1.20 (0.65, 2.23)	0.83 (0.37, 1.83)	0.78 (0.38, 1.59)	1.39 (0.73, 3.65)	0.43 (0.11, 1.65)
Captivity	3.07 (1.44, 6.55) **	2.51 (1.27, 4.93) **	2.31 (1.04, 5.10)*	2.43 (1.15, 5.12) *	4.01 (1.55, 10.37) **	0.62 (0.16, 2.39)
Life threatening illness or injury	0.83 (0.41, 1.65)	1.27 (0.71, 2.28)	0.77 (0.39, 1.53)	0.75 (0.40, 1.39)	0.87 (0.35, 2.19)	0.28 (0.11, 0.72) **
Severe human suffering	1.47 (0.76, 2.85)	1.43 (0.83, 2.47)	1.70 (0.87, 3.32)	1.21 (0.67, 2.19)	2.45 (0.91, 6.59)	1.42 (0.57, 3.56)
Sudden violent death	0.90 (0.45, 1.79)	0.85 (0.48, 1.49)	1.14 (0.56, 2.30)	0.69 (0.38, 1.27)	1.39 (0.51, 3.78)	1.19 (0.43, 3.26)
Sudden accidental death	1.31 (0.61, 2.83)	0.84 (0.46, 1.52)	1.06 (0.51, 2.21)	1.08 (0.57, 2.05)	1.40 (0.51, 3.90)	2.50 (0.76, 8.24)
Serious injury, harm, or death you caused to someone else	2.08 (0.92, 4.71)	1.69 (0.82, 3.46)	2.57 (1.13, 5.81) *	1.57 (0.70, 3.50)	2.77 (1.03, 7.44) *	2.79 (0.92, 8.45)
Any other very stressful event or experience	2.04 (1.06, 3.94) *	1.21 (0.69, 2.11)	1.62 (0.83, 3.15)	1.27 (0.68, 2.36)	0.85 (0.33, 2.18)	1.23 (0.48, 3.17)
Total number of different types of PPTE	1.13 (1.03, 1.25) **	1.05 (0.97, 1.12)	1.05 (0.96, 1.14)	1.02 (0.94, 1.10)	1.14 (1.00, 1.29)	0.95 (0.85, 1.07)

Notes. AOR = Odds ratio adjusted for sex, age, education, marital status, province; CI = Confidence Interval. AUDIT = Alcohol Use Disorder; CCG = Canadian Coast Guard; C&P = Conservation and Protection; GAD = Generalized Anxiety Disorder; MDD = Major Depressive Disorder; PTSD = Posttraumatic Stress Disorder; PD = Panic Disorder; PPTE = Potentially Psychologically Traumatic Event; SAD = Social Anxiety Disorder. * *p* ≤ 0.05; ** *p* ≤ 0.01 -Statistically significantly different.

**Table 6 ijerph-19-15116-t006:** Associations Between Potentially Psychologically Traumatic Events (PPTEs) Exposure and Screening Positive for Mental Disorders, Canadian Coast Guard (CCG) Sample (Adjusted for Sociodemographics).

Type of PPTE	PTSD	MDD	GAD	SAD	PD	AUD
	AOR (95% CI)	AOR (95% CI)	AOR (95% CI)	AOR (95% CI)	AOR (95% CI)	AOR (95% CI)
Life threatening natural disaster	1.51 (0.67, 3.41)	1.59 (0.78, 3.24)	1.08 (0.47, 2.47)	1.63 (0.77, 3.42)	3.17 (0.85, 11.85)	0.25 (0.08, 0.78) *
Fire or explosion	2.15 (0.83, 5.52)	1.47 (0.67, 3.19)	1.49 (0.56, 3.96)	1.44 (0.64, 3.25)	2.63 (0.66, 10.53)	0.31 (0.09, 1.03)
Serious transportation accident	2.56 (0.80, 8.17)	0.93 (0.39, 2.23)	0.83 (0.29, 2.35)	1.45 (0.57, 3.72)	0.93 (0.25, 3.45)	1.46 (0.27, 7.82)
Serious accident at work, home, or during recreational activity	1.98 (0.78, 5.02)	1.38 (0.63, 3.01)	1.01 (0.41, 2.50)	1.00 (0.46, 2.20)	1.59 (0.46, 5.48)	0.69 (0.22, 2.20)
Exposure to toxic substance	2.72 (1.23, 6.01) *	1.41 (0.73, 2.74)	2.12 (0.94, 4.77)	1.58 (0.77, 3.23)	1.26 (0.47, 3.38)	1.30 (0.47, 3.64)
Physical assault	1.99 (0.81, 4.90)	2.54 (1.15, 5.62) *	1.31 (0.54, 3.18)	1.08 (0.50, 2.34)	1.50 (0.48, 4.70)	0.78 (0.26, 2.35)
Assault with a weapon	2.37 (1.10, 5.13) *	1.36 (0.70, 2.62)	0.93 (0.43, 2.02)	0.95 (0.47, 1.92)	1.23 (0.46, 3.33)	0.63 (0.22, 1.81)
Sexual assault	2.40 (1.09, 5.32) *	1.61 (0.82, 3.18)	1.13 (0.51, 2.49)	0.85 (0.41, 1.74)	1.48 (0.56, 4.08)	0.97 (0.34, 2.80)
Other unwanted or uncomfortable sexual experience	1.84 (0.81, 4.16)	1.29 (0.63, 2.63)	0.79 (0.34, 1.84)	0.85 (0.40, 1.83)	0.82 (0.29, 2.33)	0.74 (0.23, 2.42)
Combat	2.78 (1.21, 6.38) *	1.33 (0.62, 2.89)	0.78 (0.28, 2.13)	0.91 (0.39, 2.11)	1.88 (0.65, 5.41)	0.51 (0.12, 2.17)
Captivity	3.65 (1.45, 9.19) **	3.57 (1.49, 8.55) **	2.73 (0.99, 7.50)	2.91 (1.15, 7.35) *	4.19 (1.43, 12.28) **	0.88 (0.21, 3.68)
Life threatening illness or injury	1.09 (0.49, 2.45)	1.38 (0.68, 2.84)	0.57 (0.25, 1.30)	0.81 (0.39, 1.69)	1.11 (0.38, 3.21)	0.26 (0.09, 0.78) *
Severe human suffering	1.70 (0.78, 3.70)	1.58 (0.80, 3.11)	1.65 (0.75, 3.66)	1.05 (0.52, 2.11)	1.66 (0.58, 4.74)	2.20 (0.71, 6.86)
Sudden violent death	1.34 (0.60, 3.02)	1.06 (0.53, 2.13)	1.09 (0.47, 2.50)	0.77 (0.38, 1.58)	1.12 (0.39, 3.22)	1.71 (0.47, 6.29)
Sudden accidental death	1.67 (0.68, 4.09)	1.05 (0.50, 2.20)	1.01 (0.42, 2.46)	1.19 (0.57, 2.51)	1.52 (0.49, 4.73)	1.99 (0.51, 7.76)
Serious injury, harm, or death you caused to someone else	1.76 (0.65, 4.80)	1.42 (0.55, 3.64)	2.37 (0.84, 6.74)	0.98 (0.34, 2.84)	2.17 (0.65, 7.20)	4.08 (1.22, 13.62) *
Any other very stressful event or experience	1.75 (0.81, 3.78)	0.83 (0.41, 1.68)	1.57 (0.71, 3.48)	0.90 (0.42, 1.92)	0.68 (0.22, 2.11)	1.02 (0.34, 3.04)
Total number of different types of PPTE	1.18 (1.06, 1.32)	1.10 (0.99, 1.20)	1.04 (0.93, 1.16)	1.02 (0.93, 1.12)	1.11 (0.97, 1.27)	0.96 (0.84, 1.11)

Notes. AOR = Odds ratio adjusted for sex, age, education, marital status, province; CI = Confidence Interval. AUDIT = Alcohol Use Disorder; CCG = Canadian Coast Guard; C&P = Conservation and Protection; GAD = Generalized Anxiety Disorder; MDD = Major Depressive Disorder; PTSD = Posttraumatic Stress Disorder; PD = Panic Disorder; PPTE = Potentially Psychologically Traumatic Event; SAD = Social Anxiety Disorder. * *p* ≤ 0.05; ** *p* ≤ 0.01 -Statistically significantly different.

**Table 7 ijerph-19-15116-t007:** Associations Between Potentially Psychologically Traumatic Events (PPTEs) Exposure and Screening Positive for Mental Disorders, Conservation and Protection (C&P) Sample (Adjusted for Sociodemographics).

Type of PPTE	PTSD	MDD	GAD	SAD	PD	AUD
	AOR (95% CI)	AOR (95% CI)	AOR (95% CI)	AOR (95% CI)	AOR (95% CI)	AOR (95% CI)
Life threatening natural disaster	1.17 (0.14, 9.59)	0.78 (0.22, 2.71)	0.64 (0.13, 3.08)	0.75 (0.18, 3.04)	-	-
Fire or explosion	0.83 (0.16, 4.31)	1.22 (0.36, 4.08)	3.04 (0.56, 16.66)	1.01 (0.27, 3.69)	-	-
Serious transportation accident	0.19 (0.02, 1.99)	0.21 (0.05, 0.93) *	4.39 (0.28, 68.35)	0.08 (0.01, 0.51) **	-	-
Serious accident at work, home, or during recreational activity	0.42 (0.07, 2.57)	0.68 (0.20, 2.32)	2.17 (0.35, 13.53)	0.44 (0.12, 1.59)	-	-
Exposure to toxic substance	3.43 (0.57, 20.56)	1.00 (0.36, 2.77)	1.07 (0.29, 3.94)	1.20 (0.36, 3.99)	-	-
Physical assault	-	1.24 (0.29, 5.39)	1.86 (0.26, 13.44)	1.25 (0.23, 6.88)	-	-
Assault with a weapon	2.97 (0.19, 45.40)	1.56 (0.44, 5.57)	0.74 (0.15, 3.64)	1.00 (0.22, 4.61)	-	-
Sexual assault	2.95 (0.46, 19.12)	0.26 (0.09, 0.72)	0.37 (0.09, 1.58)	0.81 (0.24, 2.77)	0.18 (0.02, 1.83)	-
Other unwanted or uncomfortable sexual experience	16.88 (1.31, 216.95) *	1.23 (0.45, 3.35)	0.76 (0.18, 3.19)	4.56 (0.96, 21.61)	0.67 (0.07, 6.63)	-
Combat	0.63 (0.13, 3.12)	0.99 (0.32, 3.06)	1.42 (0.28, 7.14)	0.40 (0.08, 2.01)	0.12 (0.01, 4.20)	-
Captivity	1.81 (0.36, 9.02)	1.71 (0.49, 5.98)	2.10 (0.43, 10.22)	1.81 (0.43, 7.57)	6.62 (0.46, 95.04)	-
Life threatening illness or injury	0.44 (0.09, 2.06)	1.46 (0.45, 4.76)	1.95 (0.38, 9.93)	0.60 (0.17, 2.17)	1.01 (0.09, 11.56)	-
Severe human suffering	1.02 (0.24, 4.36)	1.26 (0.46, 3.48)	1.96 (0.47, 8.22)	1.80 (0.52, 6.16)	-	-
Sudden violent death	0.16 (0.03, 0.88) *	0.58 (0.18, 1.86)	1.37 (0.25, 7.57)	0.40 (0.10, 1.60)	-	-
Sudden accidental death	0.67 (0.12, 3.88)	0.56 (0.18, 1.78)	1.36 (0.26, 7.16)	1.04 (0.25, 4.33)	3.63 (0.15, 85.12)	-
Serious injury, harm, or death you caused to someone else	3.81 (0.68, 21.31)	2.06 (0.56, 7.57)	2.41 (0.52, 11.26)	6.00 (1.17, 30.91) *	15.94 (1.01, 250.74) *	-
Any other very stressful event or experience	5.23 (0.84, 32.67)	2.97 (0.94, 9.44)	1.85 (0.43, 7.87)	2.64 (0.71, 9.89)	4.48 (0.32, 63.26)	-
Total number of different types of PPTE	1.04 (0.86, 1.26)	0.98 (0.86, 1.11)	1.06 (0.89, 1.26)	0.99 (0.86, 1.15)	1.63 (0.97, 2.73)	-

Notes. AOR = Odds ratio adjusted for sex, age, education, marital status, province; CI = Confidence Interval. AUDIT = Alcohol Use Disorder; CCG = Canadian Coast Guard; C&P = Conservation and Protection; GAD = Generalized Anxiety Disorder; MDD = Major Depressive Disorder; PTSD = Posttraumatic Stress Disorder; PD = Panic Disorder; PPTE = Potentially Psychologically Traumatic Event; SAD = Social Anxiety Disorder. * *p* ≤ 0.05; ** *p* ≤ 0.01 -Statistically significantly different.

## Data Availability

Not applicable.

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
