# Peer review of "Exposures to Potentially Psychologically Traumatic Events among Canadian Coast Guard and Conservation and Protection Officers"

_ijerph, 2022, doi:10.3390/ijerph192215116_

Round 1

Reviewer 1 Report

The topic is very interesting and relevant. The paper is written well overall and the methods seem appropriate. Content regarding p values is unclear. In the abstract, there is a "ps" value which I suspect is a typographical error. In the results tables, effect size and p values appear to be used synonymously and these are different. However, the effect size values do not align with the notated p values so perhaps it is just a confusing format. This needs to be clarified. 

Author Response

Thank you for your positive comment and feedback to strengthen the manuscript. We appreciate your time and effort spent to review the manuscript and hope you will find our updates satisfactory and the manuscript suitable for publication.

The “ps” has been used in the abstract to indicate “all p values” as in the p values for each type of PPTE listed in Table 2. We have updated the abstract to say “all ps” to provide further clarity.

In Table 1, effect size and p values were not used synonymously; instead, the presented values are effect sizes and the asterisks indicate the significance levels as defined in the note underneath the table. We modified have updated the note for more clarification as follows:

“*p < .05, **p < .01, ***p < .001– Statistically significantly different at these p-value levels”

Reviewer 2 Report

From a classical public health perspective, the proposed study is interesting and innovative:
- the theoretical part is comprehensive and the methodology well described. 
- the results are correctly presented from a methodological point of view
- the discussion, limitations and conclusions are consistent and well described.

However, given the large amount of data and rows and columns, although the results are correctly presented, the tables are unreadable and difficult to consult. I suggest that the authors reduce the text of the labels and not necessarily report all the descriptive data, but make a selection with respect to the most relevant ones (then commented on in the Discussion and Conclusions paragraphs), and highlight, in the cells of the tables, the most relevant and interesting results.

The main question addressed by the research regards the exposures to psychologically traumatic events among canadian coast guard and conservation and protection officers,

This topic is original in that it adopts a psychological perspective and relevant in that this category of workers is significantly exposed to this type of risk

In a descriptive perspective, the conclusions are consistent with the evidence and arguments presented and the authors address the main question posed

The tables contain a lot of data, the authors should try to make them more concise, and highlight the most important and significant ones, especially with respect to the discussion and conclusions sections.

Author Response

Thank you for your positive comments and provided feedback that will allow us to strengthen the manuscript. We appreciate your time and effort spent to review the manuscript and hope you will find our updates satisfactory and the manuscript suitable for publication. We agree that the tables are extensive and contain a lot of data. We have updated the tables to be more concise and easier to read. Instead of using tracked changes to update and move the entire table, I have highlighted the tables that have been updated. Some track changes were also used in the table notes. Table 1 now presents sociodemographic characteristics for the total, CCG and C&P samples. Table 2 now presents the Total Number of Different Types of Potentially Psychologically Traumatic Events (PPTEs) Exposures Across Total Sample, CCG, and C&P Officers. Table 3 has been updated to present the Prevalence of Potentially Psychologically Traumatic Events (PPTEs) Exposure Types and frequencies and comparisons between the total sample, other PSP, and the general population. An extended version of the table has also been included in the supplementary materials that displays the type of exposure (i.e., happened to me, witnesses it, learned about, part of my job).

Reviewer 3 Report

The manuscript addresses a topic that, although it has been deeply explored over the last decades, continues to arouse the interest of many students and professionals from different areas. The authors emphasize that their results “offer the first empirical evidence of potentially psychological traumatic event exposure histories” of a particular professional class, a factor that contributes to the relevance of the manuscript. However, this type of statement should be avoided, as other studies may be unknown to the authors or published before their study is. A better expression would be "to the best of our knowledge, this is the first...". In addition to this detail, the manuscript has some flaws that need to be corrected before possible publication.

  Below are a few noteworthy problems:

Abstract:

1.       when using for the first time the abbreviation PSP, the its meaning must be present (line 18).

2.       if the authors consider relevant to present % of male participants in this section, the % of female must be included.

3.       when saying that participants reported lower frequencies of lifetime exposures to PPTEs than other Canadian PSP, we wonder who the other PSPs are. Not only in the abstract, but throughout the text the authors do not specify to which group they are comparing their sample.

Introduction:

1.       Check reference numbers. For example, reference 5 appears after 6 and 7 (lines 65-68).

2.       Although the authors had presented correctional workers as PSP at the beginning of the introduction (line 39), they divided Canadian PSP and correctional workers later (line 80).

Materials and Methods

1.       When presenting sample sociodemographic characteristics, the authors repeated that the most of participants were white (lines 137 and 139).

2.       The authors recognize that self-report survey is not a way to determine a diagnosis (lines 159-161; 388-391). So, I suggest that instead of saying "disorder", the authors prefer "symptoms of x disorder"

3.       Usually, a better description of the instruments is required, as many readers may not know them. I suggest that the authors briefly present each instrument (maybe in single paragraphs), including how it is scored and the their psychometric data.

4.       When presenting statistical analyses, I believe the authors meant “performed” rather than “presented” in line 185.

5.       Authors need to clarify they analyzed CCG and C&P separately (line 192).

6.       “Prevalence of lifetime exposure to each PPTE were then compared against published PSP and general population performing a series of binomial tests”. Published PSP? I believe they meant that they compared their results with published studies on PSP and the general population.

7.       What software was used to perform the analyses?

Results:

1.       I suggest the presentation of a table with only the sociodemographic characteristics of the study samples.

2.       Tables may be presented as supplementary materials. As they are extensive and with a lot of information, in the middle of the text, the tables hinder reading.

3.       Effect size and other important test results should be presented in the text so that the reader understands their relevance.

4.       It is important to clarify who the other Canadian PSP are, mainly when comparing them with the study sample.

The presentation of the results is the part that requires the most improvement, as it is a bit confusing.

Discussion:

1.       While in line 274 the authors said that “7.5% of the current sample reported never being exposed to a PPTE”, in lines 280-281they presented that 99.7% reported have been exposed to PPTE.

2.       In the sentence "Similar PPTEs 290 were reported by the general population and CCG/C&P members, the rates 291 of exposure were higher for CCG and C&P members", an adverb is missing.

3.       The authors said that “only 37.4% of the sample was female” (line 350). However, they previously said that 55.2% (abstract) /55.3% (materials and methods) of the sample was male. Please check all the numbers presented throughout the manuscript.

4.       "n% of the sample was" not "were". Please check grammar rules.

I hope these comments are a useful guide for authors to improve the manuscript.

Author Response

The manuscript addresses a topic that, although it has been deeply explored over the last decades, continues to arouse the interest of many students and professionals from different areas. The authors emphasize that their results “offer the first empirical evidence of potentially psychological traumatic event exposure histories” of a particular professional class, a factor that contributes to the relevance of the manuscript. However, this type of statement should be avoided, as other studies may be unknown to the authors or published before their study is. A better expression would be "to the best of our knowledge, this is the first...". In addition to this detail, the manuscript has some flaws that need to be corrected before possible publication.

Thank you for your comments and feedback to strengthen the manuscript. We appreciate your time and effort spent to review the manuscript and hope you will find our updates satisfactory and the manuscript suitable for publication.

We agree that the statement you have highlighted should be changed to be more inclusive of all research. We have updated the statement as you have suggested in the following locations in the manuscript (abstract, line 7; Discussion, line 270; line 406; line 414). We have also addressed your other comments below.

  Below are a few noteworthy problems:

Abstract:

  1. when using for the first time the abbreviation PSP, the its meaning must be present (line 18).

Thank you for identifying this. It has been corrected.

  1. if the authors consider relevant to present % of male participants in this section, the % of female must be included.

Thank you for this suggestion. We have also added the % of females to the abstract.

  1. when saying that participants reported lower frequencies of lifetime exposures to PPTEs than other Canadian PSP, we wonder who the other PSPs are. Not only in the abstract, but throughout the text the authors do not specify to which group they are comparing their sample.

Thank you for identifying this opportunity to clarify in the manuscript. We have updated the manuscript to identify the “other Canadian PSP” (i.e., municipal/provincial police, RCMP, correctional workers, dispatchers, firefighters, paramedics).

Introduction:

  1. Check reference numbers. For example, reference 5 appears after 6 and 7 (lines 65-68).

Thank you for identifying this error. It has been corrected.

  1. Although the authors had presented correctional workers as PSP at the beginning of the introduction (line 39), they divided Canadian PSP and correctional workers later (line 80).

Thank you for identifying this area of confusion. We have removed the words correctional workers and cited both papers after Canadian PSP as correctional workers are Canadian PSP.

Materials and Methods

  1. When presenting sample sociodemographic characteristics, the authors repeated that the most of participants were white (lines 137 and 139).

Duplication removed.

  1. The authors recognize that self-report survey is not a way to determine a diagnosis (lines 159-161; 388-391). So, I suggest that instead of saying "disorder", the authors prefer "symptoms of x disorder"

Thank you for identifying this opportunity to clarify. We have added to the paragraph to clarify what we are referring to when we say positive screening. We have kept the original language to be consistent with our previously published papers and the Public Health Agency of Canada.

  1. Usually, a better description of the instruments is required, as many readers may not know them. I suggest that the authors briefly present each instrument (maybe in single paragraphs), including how it is scored and the their psychometric data.

Thank you for this suggestion. We have updated the methods section to include expanded information about each of the measures and information on their reliability and internal consistency.

  1. When presenting statistical analyses, I believe the authors meant “performed” rather than “presented” in line 185.

Corrected.

  1. Authors need to clarify they analyzed CCG and C&P separately (line 192).

This information has been added.

  1. “Prevalence of lifetime exposure to each PPTE were then compared against published PSP and general population performing a series of binomial tests”. Published PSP? I believe they meant that they compared their results with published studies on PSP and the general population.

Yes, thank you for catching this. This has been updated to be more clear.

  1. What software was used to perform the analyses?

This information has been added to the statistical analyses paragraph.

Results:

  1. I suggest the presentation of a table with only the sociodemographic characteristics of the study samples.

Thank you for this suggestion. Table 1 has been updated to be sociodemographic characteristics only. Table 2 now displays only displays data on the Total Number of Different Types of Potentially Psychologically Traumatic Events (PPTEs) Exposures Across Total Sample, CCG, and C&P Officers.

  1. Tables may be presented as supplementary materials. As they are extensive and with a lot of information, in the middle of the text, the tables hinder reading.

Thank you for this suggestion. We have updated Tables 1-3 to be more concise. An expanded version of Table 3 is now available as supplementary material. We also moved the position of the tables. Table 1 now appears after participant information while Tables 2-7 now appear at the end of the manuscript. We hope this makes the results easier to read.

  1. Effect size and other important test results should be presented in the text so that the reader understands their relevance.

Thank you for this suggestion. We have added effect size and other statistical information to the text in the results section.

  1. It is important to clarify who the other Canadian PSP are, mainly when comparing them with the study sample.

Thank you for identifying this opportunity to clarify in the manuscript. We have updated the manuscript to identify the “other Canadian PSP” (i.e., municipal/provincial police, RCMP, correctional workers, dispatchers, firefighters, paramedics).

The presentation of the results is the part that requires the most improvement, as it is a bit confusing.

The results section has been updated with additional statistical information and the tables have been removed and moved to the end of the manuscript.

Discussion:

  1. While in line 274 the authors said that “7.5% of the current sample reported never being exposed to a PPTE”, in lines 280-281they presented that 99.7% reported have been exposed to PPTE.

                        Thank you for identifying this mistake. The text has been updated as this was an error. 7.5% reported never being exposed. 92.5% reported exposure to at least one PPTE.

  1. In the sentence "Similar PPTEs 290 were reported by the general population and CCG/C&P members, the rates 291 of exposure were higher for CCG and C&P members", an adverb is missing.

This has been corrected.

  1. The authors said that “only 37.4% of the sample was female” (line 350). However, they previously said that 55.2% (abstract) /55.3% (materials and methods) of the sample was male. Please check all the numbers presented throughout the manuscript.

                        Thank you for identifying this opportunity to clarify. The total percentages may not sum to 100 and ns may not sum to 412 due to non-response. This information has been added under Table 1.

  1. "n% of the sample was" not "were". Please check grammar rules.

We have updated our grammar where appropriate.

I hope these comments are a useful guide for authors to improve the manuscript.